# Probable Mechanisms of Doxorubicin Antitumor Activity Enhancement by Ginsenoside Rh2

**DOI:** 10.3390/molecules27030628

**Published:** 2022-01-19

**Authors:** Alexander Popov, Anna Klimovich, Olga Styshova, Alexander Tsybulsky, Dmitry Hushpulian, Andrey Osipyants, Anna Khristichenko, Sergey Kazakov, Manuj Ahuja, Navneet Kaidery, Bobby Thomas, Vladimir Tishkov, Abraham Brown, Irina Gazaryan, Andrey Poloznikov

**Affiliations:** 1G.B. Elyakov Pacific Institute of Bioorganic Chemistry, Far Eastern Branch, Russian Academy of Sciences, 159 Prospect 100-Years of Vladivostok, 690022 Vladivostok, Russia; krivoshapkoon@mail.ru; 2Department of Biochemistry, Microbiology and Biotechnology, Institute of the World Ocean, Far Eastern Federal University, Campus L, Island Russian, 690920 Vladivostok, Russia; avt_biotech@mail.ru (A.T.); hushpulian@gmail.com (D.H.); uchsovet@dvfu.ru (A.O.); 3Faculty of Biology and Biotechnology, National Research University Higher School of Economics, 13-4 Myasnitskaya Street, 117997 Moscow, Russia; igazaryan@gmail.com (I.G.); andrey.poloznikov@gmail.com (A.P.); 4Moscow Institute of Physics and Technology, 9 Institutskiy lane, Dolgoprudny, 141701 Moscow, Russia; hristik14@gmail.com; 5Department of Chemistry and Physical Sciences, Dyson College of Art and Sciences, Pace University, 861 Bedford Road, Pleasantville, NJ 10570, USA; skazakov@pace.edu; 6Departments of Pediatrics, Darby Research Institute, Neurosciencec, Drug Discovery, Medical University of South Carolina, Charleston, SC 29425, USA; manujahu@buffalo.edu (M.A.); ammalkai@musc.edu (N.K.); thomasbo@musc.edu (B.T.); 7Innovation and High Technologies MSU Ltd., Tsymlyanskaya 16, 109599 Moscow, Russia; vitishkov@gmail.com; 8Department of Chemical Enzymology, School of Chemistry, M.V. Lomonosov Moscow State University, 1 Leninskiye Gory, 119991 Moscow, Russia; 9Bach Institute of Biochemistry, Research Center of Biotechnology of the Russian Academy of Sciences, 33-2 Leninsky Prospect, 119071 Moscow, Russia; 10Department of Anatomy Cell Biology, New York Medical College, 15 Dana Road, Valhalla, NY 10595, USA; abraham_brown@nymc.edu

**Keywords:** ginsenosides, Rh2, cancer therapy, ROS, Nrf2, reporter assay, computer modeling

## Abstract

Ginsenoside Rh2 increases the efficacy of doxorubicin (DOX) treatment in murine models of solid and ascites Ehrlich’s adenocarcinoma. In a solid tumor model (treatment commencing 7 days after inoculation), DOX + Rh2 co-treatment was significantly more efficacious than DOX alone. If treatment was started 24 h after inoculation, the inhibition of tumor growth of a solid tumor for the DOX + Rh2 co-treatment group was complete. Furthermore, survival in the ascites model was dramatically higher for the DOX + Rh2 co-treatment group than for DOX alone. Mechanisms underlying the combined DOX and Rh2 effects were studied in primary Ehrlich’s adenocarcinoma-derived cells and healthy mice’s splenocytes. Despite the previously established Rh2 pro-oxidant activity, DOX + Rh2 co-treatment revealed no increase in ROS compared to DOX treatment alone. However, DOX + Rh2 treatment was more effective in suppressing Ehrlich adenocarcinoma cell adhesion than either treatment alone. We hypothesize that the benefits of DOX + Rh2 combination treatment are due to the suppression of tumor cell attachment/invasion that might be effective in preventing metastatic spread of tumor cells. Ginsenoside Rh2 was found to be a modest activator in a Neh2-luc reporter assay, suggesting that Rh2 can activate the Nrf2-driven antioxidant program. Rh2-induced direct activation of Nrf2 might provide additional benefits by minimizing DOX toxicity towards non-cancerous cells.

## 1. Introduction

Oncological diseases are widespread, multifaceted and, hence, difficult to cure. Numerous chemotherapeutic medicines are used clinically, but none can be referred to as an effective, universal and harmless treatment; a “magic bullet” does not exist. One of the most effective and widely used antitumor agents is the anthracycline doxorubicin (DOX). DOX is widely used as monotherapy and in various combinations for cancer treatment. The antitumor activity of DOX is thought to be based on its genotoxicity, i.e., DOX inhibits the progression of topoisomerase II by intercalating DNA and on its ability to generate reactive oxygen species (ROS). The major side effects of DOX are carditoxicity, neuropathy, hepatotoxicity, nephrotoxicity, amyelosuppression, neutropenia, anaemia among others, which are caused mainly due to the high cytotoxicity to both tumor and many normal cells [1,2].

One solution to the problem of insufficient efficacy and unwanted side effects of DOX could be its co-administration with another medication that exhibits either complementary antitumor activity or modulates various physiological (immune, nervous, and hormonal) responses. A combinatorial approach might thus increase the efficiency of DOX action and decrease the severity of side effects. Our preliminary testing of various naturally occurring adjuvants to DOX revealed ginsenoside Rh2 as a promising agent [3,4]. Ginsenoside Rh2 belongs to the group of triterpene glycosides found exclusively in the legendary *Panax ginseng* root. Despite the determination that ginseng contains only traces of ginsenoside Rh2, it has been reported that Rh2 exhibits the widest medico-biological activity and chemotherapeutic efficacy among all other ginsenosides [5,6,7]. The effectiveness observed for ginseng itself may originate from the fact that the majority of natural ginseng glycosides contain the 20S-protopanaxadiol group, which is converted into Rh2 and its genin in stomach and intestines [4], as a result of acidic and enzymatic hydrolysis. It has been established that treatment with ginsenoside Rh2 results in apoptosis, necrosis, and/or autophagy of cancer cells [8,9].

In this study, we demonstrate an improved effectiveness of DOX + Rh2 combination compared to their individual action in a solid tumor model of Ehrlich’s adenocarcinoma. The mode of action for Rh2 was probed in vivo by utilizing early treatment of solid-tumor and ascites tumor models. Mechanistic insights into Rh2 biological action were also gained by using several cell-based molecular approaches: (a) mild pro-oxidant activity of Rh2 on tumor (Ehrlich’s adenocarcinoma) and immune (splenocytes) cells was confirmed using the cell-permeable fluorescent ROS indicator, 2′,7′-dichlorofluorescein diacetate (DCFDA); (b) stabilization of Nrf2 transcription factor, the master regulator of the genetic antioxidant response, by micromolar Rh2, was shown using a Neh2-luc fusion reporter assay developed by the authors; whereas DOX, contrary to lapachone, a known ROS generator and potent Neh2-luc reporter activator [10], did not activate the reporter pointing to DOX-induced ROS production as a delayed side effect of DOX action on nuclear DNA; (c) docking studies supported the possibility of Rh2 direct interaction with Keap1 BTB-domain by analogy with another Nrf2 activator of triterpenoid nature, bardoxolone; (d) in vitro cell adhesion test pointed to the additive character of DOX + Rh2 effect on cellular attachment and invasion.

## 2. Results

### 2.1. Enhanced Anti-Tumor Effects of DOX and Rh2 in a Solid Tumor Model

#### 2.1.1. Solid Tumor Model—Delayed Treatment (Variant A, Post-Tumor Formation)

Visual comparison of tumor sizes on day 22 clearly shows (Figure 1A) that DOX alone has a definite inhibitory effect on tumor growth, and this effect is enhanced by its combination with Rh2, whereas Rh2 has no effect. A graphical presentation of averaged tumor volumes at different times post-inoculation (Figure 1B), demonstrates that drug treatments resulted in statistically significant benefits. Univariate analysis of variance indicates a significant DOX-Rh2 interaction (*p* < 0.02). Post hoc analysis indicates pairwise significance between DOX alone and DOX + Rh2 on days 13–22 (*p* < 0.02), suggesting a non-additive (synergistic) treatment effect for Rh2 with DOX.

As seen in Figure 1A, tumor growth accelerates after day 13 and the relative inhibition of drug treatment diminishes somewhat. On day 13, DOX alone inhibits tumor growth by about 20%, whereas DOX + Rh2 results in almost 40% inhibition. By day 13, inhibition by DOX treatment is less than 20% and DOX + Rh2 is only approximately 30%. 

In order to more fully explore the diminishing benefit of combination treatment at later times, we compared the traditional model just presented with protocols in which treatment is initiated sooner after tumor inoculation. The sections below present the effects of DOX and Rh2 treatment in an early treatment solid tumor model (Variant B) and in the ascites liquid tumor model.

#### 2.1.2. Solid Tumor Model—Early Treatment (Variant B; 24 h after Inoculation of Tumor Cells)

Combination (DOX + Rh2) treatment is exceptionally beneficial when DOX and Rh2 are combined (Figure 2), as no tumor formation is observed. Although Rh2 treatment alone delays tumor formation, by day 13 after inoculation, all animals have formed a measurable tumor (Figure 2). In the DOX-treated group, only four out of seven animals developed a tumor. On day 17, the average tumor weights were 0.2756 ± 0.174 g (N group, seven animals), 0.0647 ± 0.029 g (Rh2 group, seven animals), and 0.0198 ± 0.024 g (DOX group, four animals had tumor) (typical tumors shown in Figure 3). 

A comparison of tumor size/weight showed that both monotherapies, e.g., Rh2 and DOX, inhibit tumor growth compared to the group of untreated animals (group N). However, in the Rh2 group, all animals developed a tumor, whereas in DOX group, some animals (three out of seven) did not develop a tumor by day 17. In contrast, the combined action of both drugs resulted in the complete suppression of tumor formation. 

### 2.2. Enhanced Anti-Tumor Effects of DOX and Rh2 in the Ascites Model

The Kaplan–Meier survival plot for the four groups of intraperitoneal tumor-inoculated animals is shown in Figure 4. The graphical results indicate that either DOX or ginsenoside Rh2 alone exhibits significant anti-tumor activity and prolong the survival of tumor-inoculated animals (Figure 4). Indeed, the median survival of the untreated control group was 21 d, while that of the DOX-treated group was 65 d and Rh2-treated group was 60 d. *p*-values were calculated in comparison to Group A. In contrast, the co-treatment with DOX + Rh2 resulted in better survival than the individual treatment with DOX or Rh2 alone. The median survival of DOX + Rh2 combination-treated group (Group D) was undefined, since 88% of the animals were alive at the end of the 100-day observation period (*p* < 0.001, log-rank (Mantel-Cox), df = 1, χ^2^ = 15.925). Although DOX-treated animals tended to survive longer than Rh2-treated ones, the difference between DOX and Rh2 was not statistically significant (*p* = 0.626, log-rank test, df = 1, χ^2^ = 0.237). 

The improvement in DOX + Rh2 survival was statistically significant in comparison to the monotherapy groups: DOX treatment (*p* = 0.043, log-rank test, df = 1, χ^2^ = 4.091) and Rh2 (*p* = 0.013, log-rank test, df = 1, χ^2^ = 6.186). Thus, co-treatment with DOX and Rh2 display synergistic action. This suggests that the introduction of Rh2 as an adjuvant for DOX therapy could be extremely beneficial in terms of survival, at least if it can be introduced soon after tumor induction. In the sections below, we used assays in cell culture models to explore the mechanisms underlying the enhanced activity of DOX + Rh2 in survival enhancement in vivo.

### 2.3. Effect of Rh2 on ROS Production in Primary Cell Cultures of Adenocarcinoma and Splenocytes

Our first hypothesis to explain the observed enhancement in the combined DOX + Rh2 anti-tumor action was based on their documented individual pro-oxidant activities, which is detrimental for cancer cells and activating for immune cells. DOX generates reactive oxygen species (ROS) [1]. The ROS-modulating activity of ginsenosides is well known [11] and has been structurally characterized [12]; Rh2, in particular, behaves as a pro-oxidant. We addressed the question of whether Rh2 co-administration changes ROS production in Ehrlich’s adenocarcinoma cells compared to DOX-treated splenocytes derived from healthy mice. 

ROS production (H_2_O_2_) was assessed with a cell-permeable fluorescent indicator, DCFDA. As shown in Figure 5A, DOX treatment results in a two-fold increase in fluorescence when compared to non-treated preparations of adenocarcinoma cells. Ginsenoside Rh2 exerts a similar pro-oxidant effect on the adenocarcinoma cells: it increases fluorescence two-fold if added at 1.5 μM concentration and 2.5-fold at 15 μM. However, DOX + Rh2 co-treatment results in no further increase in fluorescence. Instead, a slight decrease in fluorescence is observed (Figure 5A). Hence, the enhanced anti-tumor effect of DOX + Rh2 co-treatment cannot be ascribed to the increased ROS production upon their co-administration.

The same ROS induction experiments were repeated for splenocytes (Figure 5B) It is noteworthy that the cell density of adenocarcinoma cells (panel A) is twice that of the splenocytes (panel B). However, baseline ROS production in the adenocarcinoma cells was about one third baseline production in splenocytes. Thus, immune cells exhibit a substantially higher baseline level of ROS production that in adenocarcinoma cells. 

In splenocytes, we observe that DOX has no effect on ROS production (Figure 5B). Addition of Rh2 (15 μM) almost doubles the ROS level in the immune cells. However, the combined treatment with DOX + Rh2 results in partial quenching of Rh2-induced ROS. Overall, only Rh2 behaves as a mild pro-oxidant agent toward both adenocarcinoma and immune cells. Based upon the data in Figure 5, ROS production is an unlikely mechanism in play for the enhanced anti-tumor effect of DOX + Rh2 combination.

### 2.4. Effect of DOX + Rh2 Co-Treatment on Primary Adenocarcinoma Cell Adhesion 

We observed substantial efficacy for the DOX + Rh2 combination in the solid tumor model (variant B) with early treatment, but only a modest effect of DOX + Rh2 in a delayed treatment (variant A) of the solid tumor model. This suggested that co-treatment may compromise the ability of the adenocarcinoma cells to seed a tumor, i.e., spontaneous attachment or tissue invasion. To evaluate the efficiency of cell attachment to the surface and subsequent growth, we plated cells into microplate wells containing medium with their two-fold serial dilutions of either individual drugs or DOX + Rh2 in the 1–50 µM concentration range. After 24 h incubation, dead cells were washed out and live cells quantified visually and spectrophotometrically (Figure 6). The DOX + Rh2 treatment showed clear and statistically significant benefit over the individual treatment regimes. Thus, DOX + Rh2 combination inhibits cell adhesion much more strongly than either treatment alone, and this observation may explain the mechanism behind the efficacy of the combined treatment. 

### 2.5. Evaluation of DOX and Rh2 Activity in Neh2-luc Reporter Assay 

Many known Nrf2 activators are ROS producers. Keap1 protein is a redox sensor, and it has a thiol-disulfide regulatory switch that specifically responds to ROS and hydrogen peroxide in particular [13]. The Neh2-luc reporter provides an immediate response to drugs that generate ROS directly, immediately upon their entry into the cell. Therefore, we compared a known ROS generator which has been proposed as a cancer treatment, beta-lapachone, with DOX in the Neh2-luc reporter assay (Figure 7A). Lapachone efficiently generates superoxide which quickly dismutates to hydrogen peroxide [10]. As seen in Figure 7A, beta-lapachone is a potent Nrf2 activator. Unexpectedly, DOX is silent in the assay, suggesting that superoxide/hydrogen peroxide production does not occur as a direct (or immediate) effect of DOX introduction into the cell. This suggests that ROS production is not the originating event in DOX toxicity, but a side effect of DOX damage to nuclear DNA leading to cell death. 

Ginseng extract and its components are known to activate Nrf2, although the exact mechanism of activation is still controversial [14,15,16,17,18,19]. Ginsenosides Rb1, Rg1, and the in vivo metabolite 20S-protopanaxatriol act as mild Nrf2 activators providing 1.5–2.5-fold activation in ARE-luc reporter assay upon 24 h incubation. Activation was confirmed by RT-PCR for HO-1 and NQO-1 Nrf2-target genes, as well as Nrf2 itself [14]. Examination of the ability of Rh2 to activate Nrf2 has not been reported.

As shown in Figure 7B, Rh2 is a dose-dependent activator of the Neh2-luc reporter, providing a two-fold stabilization of the Neh2-luc fusion protein after 3 h incubation. The effect is specific for the Nrf2 reporter system, since no activation is observed for HIF1 ODD-luc reporter used as a control for assay specificity, ruling out the common effects such as inhibition of ubiquitin activating enzyme [20] or proteasome [21] in the range of Rh2 concentrations used. Since no activation effect was observed for 20S-protopanaxadiol in the reporter system at 3 h incubation (results not shown), one may expect that the glucopyranoside ring in Rh2 works toward an improved interaction with Keap1. 

Activation of the Neh2-luc reporter by Rh2, but not by DOX, may play a key role in reducing side effects due to chemotherapy. It is likely that Rh2, by stimulating Nrf2 activity, is able to provide the adaptation and sustainability of normal cells to various stressors, including oxidative stress, since Nrf2 is the main transcription factor controlling intracellular redox balance by enhancing the expression of antioxidant enzyme genes. 

### 2.6. HMOX1 Expression Induction by Rh2 and 20S-Protopanaxadiol 

In order to further substantiate that Rh2 activates the Nrf2 pathway, we examined the ability of Rh2 to stimulate expression of one of the major Nrf2 targets, heme oxygenase 1 (HMOX 1). The RT-PCR comparative analysis (Figure 8) for Rh2 and 20S-protopanaxadiol with respect to HMOX 1, demonstrates that Rh2 is a much more potent Nrf2 activator than 20S-protopanaxadiol, particularly at the 5 h time point, which is in agreement with the 3 h reporter activation assay.

In contrast, 20S-protopanaxadiol does not activate mRNA expression at 5 h. This observation agrees with the Neh2-luc reporter activation data and may reflect the importance the glucopyranoside ring for Keap1 binding. 

### 2.7. Plausible Interaction of Rh2 with BTB-Domain of Keap 1

Since Rh2 is a triterpenoid, and yet another triterpenoid, bardoxolone, is a well-characterized and extremely potent Nrf2 activator, we decided to use the recently deposited crystal structure of Keap1 BTB-domain with the co-crystallized covalently bound bardoxolone (4CXT.pdb) as a template for Rh2 docking. Bardoxolone covalently binds Cys151 in Keap1 BTB-domain, displaces Cul3 ubiquitin ligase from its complex with Keap1, thus leading to Nrf2 protein stabilization. To validate the docking procedure, bardoxolone was docked to the drug-free Keap1 BTB-domain: the resulting docking position (Figure 9A) overlapped with that in the crystal structure of the drug bound BTB-domain, thus confirming the correctness of the docking approach. The same docking protocol was then used for Rh2, to generate the model shown in Figure 9B. The C-docker interaction energy for Rh2 was close to that for bardoxolone, −34.12 and −31.14 kcal/mol, respectively, for the positions shown in Figure 9A,B, meaning that both molecules have equal potencies for non-covalent interaction with Cys151-contaning site of the BTB-domain. Glucopyranoside rings in Rb1 and Rg1 interfere with their binding into the bardoxolone site in the BTB-domain (not shown) and, hence, their very small effect in ARE-luc assay compared to that of 20S-protopanaxatriol [12] is in agreement with the docking predictions. The glucopyranoside ring in Rh2 does not prevent docking of Rh2 into the bardoxolone binding site; on the contrary, it permits a number of additional ionic interactions with Keap1 to be formed: the C-docker energy for 20S-protopanaxadiol is two-fold worse than that for Rh2. Additional modeling studies performed with LibDock program (results not shown) also places Rh2 ahead of 20S-protopanaxadiol for Keap1 binding. Hence, the glucopyranoside ring in Rh2 should play a role in Keap1 binding and Nrf2 activation.

The success of our docking procedure indicates that it is likely that Rh2 is a direct activator of Nrf2 and binds the same site as bardoxolone with similar affinity. However, Rh2 will be a much weaker Nrf2 activator since it will not form a covalent bond to Cys151 as readily as bardoxolone, which has an extreme alkylating potency and works in the nanomolar range of concentrations in Neh2-luc assay [22].

## 3. Discussion

It has been proposed that that the mechanism of antitumor action of Rh2 is due to its interaction with plasma membrane lipids with lowered cholesterol content and also rafts and caveolae [23]. Lipid rafts and reduced cholesterol content in the plasma membrane play key roles in cancer cell migration, adhesion and invasive growth, as various agents disrupting lipid rafts exert antitumor effects [24,25]. Although lipid raft disruption induced by Rh2 correlates inversely with the cholesterol content in the membrane, as confirmed for artificial [26,27,28] and biological membranes [29], Rh2 effect cannot be reversed by additional content cholesterol in lipid rafts [23]. Rh2 has high affinity for sphingomyelin, which is also one of the main components of lipid rafts [29]. Therefore, it is highly likely that sphingomyelin is among the targets for Rh2 in lipid rafts of plasma membranes. Disintegration of lipid rafts play positive roles in fighting tumor growth, for example, by inducing Fas oligomerization and apoptosis [30,31,32], or by causing disorder in CD44 localization and thus inhibiting its function in tumor cell migration [33]. 

Second, cancer cells are known to activate the expression of p-glycoprotein, which counteracts DOX penetration and results in the acquired drug resistance. Ginsenoside Rh2 has been shown to inhibit p-glycoprotein [34,35] and, moreover, to inhibit its expression [36], and in this way prevent the development of drug resistance. This mode of Rh2 action may explain the observed effect on tumor shrinking for DOX + Rh2 treatment group in a solid tumor model with delayed treatment. Third, the efficiency of combined action of DOX and Rh2 in both ascites model and an early treatment variant of the solid tumor model, as well as in the cell adhesion assay, points to inhibition of tumor cell attachment. Two Rh2 targets are directly relevant to the tumor’s ability to invade, annexin A2 (ANXA2) [37], and matrix metalloproteinase (MMP) [38]. Among all members of the MMP gene family, MMP-2 and MMP-9 are considered to be especially important in the degradation of the extracellular matrix that is associated with malignant behavior in a variety of tumor cells. Their induction was shown to be inhibited by Rh2 [39]. Annexin A2 is known to be highly expressed in the cancer cells, and its expression level is directly linked to their proliferation and invasion ability [40]. Ginsenosides Rg5 and Rk1 have minor structural difference compared to Rh2 and were also shown to bind ANXA2 [41]. Co-treatment with annexin A2 inhibitors such as matrine [42], LGRFYAASG peptide [43], or siRNA to annexin A2 [44] has been shown to improve outcomes from various cancer scenarios. In addition to these established targets, Rh2 somehow prevents p53 degradation [45], inhibits expression of AP-1 [46,47], and inactivates Akt [23,47]. All these molecular mechanisms may contribute to the enhanced anti-tumor effect of DOX + Rh2 co-treatment as depicted in Figure 10.

In immune cells, in addition to lipid raft internalization and the changing in redox-balance of cells, Rh2 may exert more specific effects [48]. Immune response in cancer is tightly bound to tumor-associated macrophages (TAM) presented by two subsets [49]: the M1 subset inhibits cell proliferation and causes tissue damage while the M2 subset promotes cell proliferation, invasion and metastasis [50,51]. Ginsenoside Rh2 is shown to convert TAMs from the M2 to the M1 subset and prevent lung cancer cell migration [52]. The immunosuppressive M2 phenotype is caused by the purinergic pathway that directs the release of extracellular ATP and its conversion to immunosuppressive adenosine as shown recently in [53]. One of the key players in purine uptake is P2X7 receptor predominantly expressed in cells of hematopoietic lineage, including macrophages, lymphocytes and microglia [54]. The growth of experimental tumors is strongly inhibited by targeting P2X7, the ATP-selective receptor of cancer and immune cells [55]. Therefore, it is most likely that the molecular mechanism of TAM conversion from the M2 to the M1 subset by Rh2 treatment directly originates from the recently discovered ability of Rh2, along with ginsenosides Rb1, Rd, and metabolite K, to allosterically modulate the P2X7 receptor [56].

Moreover, in normal cells, Rh2 contributes to the activation of crucial signaling pathways, providing adaptation to various stressors by direct action on transcription factors or redox-balance. The major transcription factor controlling intracellular redox balance is Nrf2: the level of antioxidant enzymes and reduced glutathione in the cancer cell is considerably higher than that in healthy cells due to up-regulation of the Nrf2-induced program, which contributes to cancer resistance to chemotherapy [57,58]. Hence, healthy cells are less protected from ROS-producing or alkylating antitumor agents. Inhibiting Nrf2 at the level of a whole organism as a co-treatment strategy will only increase the vulnerability of healthy cells to the treatment. On the contrary, co-treatment or pretreatment with Nrf2 activators will spare healthy cells from damage induced by antitumor agents such as DOX without compromising its effect on the cancer cell [59,60]. The observed direct activation of Neh2-luc reporter assay (Figure 7) in combination with RT-PCR (Figure 5) and modeling studies (Figure 9), points to Rh2 ability to act as a direct Nrf2 stabilizer. In fact, Rh2 is known to activate Nrf2- driven program (66). Hence, its co-administration with the established anti-tumor agents may significantly decrease harmful side effects and increase survival.

Our results and the available literature on Rh2 biological activity indicate that the enhanced effect of DOX + Rh2 co-treatment may have numerous mechanisms originating from Rh2 pleiotropic action. The benefits of DOX + Rh2 combined treatment may come from: (1) complementary action of Rh2 on cancerous cells to promote cell death pathways and/or inhibit tumor invasion, (2) Rh2-induced activation of immune response to combat cancer; (3) Rh2-induced protection against DOX side effects for healthy cells. 

It should be noted that the results obtained in this work correlate well with our earlier ideas about the characteristic features in the primary action of Rh2 on cell plasma membranes with different architectonics and network functionality, for example, tumor and immune cells [5,6,26,61,62]. 

Thus, with respect to organism response to DOX + Rh2 co-treatment, the benefits may come from the enhanced cytotoxicity towards the cancer cells, an improved immune response and enhanced general protection of healthy cells. 

## 4. Materials and Methods

### 4.1. Reagents

Ginsenoside Rh2—D-3β-O–glucopyranoside-20(S)-protopanaxadiol identical with natural ginsenoside-Rh2 (Molecular mass 622.87 Da; purity > 98%) was synthesized from betulafolientriol, in accord with the previously developed protocol [63], and kindly provided by Dr. L. N. Atopkina Pacific Institute of Bioorganic Chemistry, Far Eastern Branch, Russian Academy of Sciences. “Doxorubicin-Teva” was bought from Pharmachemie (The Netherlands), Ophtan Dexametazon—from Santen (Finland), *E. coli* endotoxin (LPS), thiobarbituric acid, 2′,7′-dichlorofluorescein diacetate (DCFDA), 20S-protopanaxadiol, beta-lapachone, were purchased from Sigma-Aldrich (St. Louis, MO, USA), dimethylsulfoxide (DMSO) from ChimMed (Moscow, Russia). All reagents were of the highest purity available and used without further purification.

### 4.2. HIF1 ODD-luc and Neh2-luc Reporter Assays

Neuroblastoma SH-SY5Y cell lines stably expressing firefly luciferase fusion reporters developed, optimized and described in [64,65] were used to study the effect of Rh2 on stabilization of HIF1 and Nrf2 transcription factor proteins in the form of their recognition domain fusions with luciferase (ODD and Neh2 domains fused to luciferase, respectively). Luminescence signal corresponds to the steady-state concentration of the fusion protein (equilibrium between fusion protein production and degradation through the recognition step, ubiquitination, and proteasomal degradation). Under the optimized conditions of the reporter assay, the rate-limiting step is controlled by the recognition step, i.e., HIF prolyl hydroxylation step for HIF1 ODD-luc reporter, and Neh2-Keap1 dissociation step for Neh2-luc reporter, respectively [65,66]. Both reporters have been successfully used for drug screening and optimization purposes: HIF1 ODD-luc reporter activation is observed in response to HIF prolyl hydroxylase inhibitors, whereas Neh2-luc reporter activation is observed in response to Nrf2 activators working by Keap1 alkylation mechanisms or Nrf2 displacement mechanisms [65,67].

Cells were plated into 96-well white flat-bottom plates (25,000 cells/well) in 100 µL DMEM/F12 medium (Glutamax, Gibco, Thermo Fisher Scientific, Waltham, MA, USA) supplemented with 10% FBS and 100 µM penicillin/streptomycin mix. Cells were incubated overnight in a CO_2_-incubator at 37 °C. Ginsenoside Rh2 was added into the medium in the 2–80 µM final concentration range from 50x stocks in DMSO (2 µL per well). Beta-lapachone (20 mM stock in DMSO), and Doxorubicin (10 mM stock in DMSO) were added to the culture media in the 1–40 µM final concentration range. Cells were incubated with drugs for 3–4 h, lysed for 7 min on a shaker and assayed with a luciferase reagent (Promega, Madison, WI, USA) on a SpectraMax M5e spectrophotometer (Molecular Devices, San Jose, CA, USA). Beta-lapachone was used as positive controls for Neh2-luc reporter line. Experiments were performed in triplicate; luminescence signal was normalized to background (2 µL pure DMSO added).

### 4.3. RT-PCR Analysis

The RT-PCR protocol used was the same as previously employed [68]. N2a mouse neuroblastoma cells were treated with either vehicle (DMSO), or ginsenosides, Rh2 or 20S-protopanaxadiol (25 µM, aliquot in DMSO), for a fixed time (5 h or 24 h). Cellular lysates were used to isolate total mRNA using TRI reagent according to the manufacturer’s protocol (Sigma-Aldrich, MO, USA). Reverse transcription of total RNA was performed using a High-Capacity cDNA Reverse Transcription Kit (Thermo Fisher Scientific, Waltham, MO, USA). The cDNA was diluted, and about 20 ng was used to amplify in an ABI prism 7900 HT Real-time PCR system (Applied Biosystems, Waltham, MA, USA) for Nrf2 target gene, heme oxygenase 1 (HMOX-1), and glyceraldehyde-3-phosphate dehydrogenase (GAPDH), a house-keeping gene, using PowerUp SYBR Green Master Mix (Thermo Fisher Scientific., Waltham, MA, USA). Cycling parameters were set as 95 °C for 10 s, followed by 60 °C for 1 min. Relative expression was calculated using the ΔΔCt method of Livak and Schmittgen. Measurements were performed in triplicate. Values are expressed as a fold change from the control reaction and normalized to GAPDH expression. Mean ± SEM is plotted as bar graph.

### 4.4. Fluorescent Assay for ROS in the Cell Culture

Primary cells of Ehrlich’s adenocarcinoma were produced from the ascites liquid of tumor-bearing mice; splenocytes were purified from the spleen of healthy CD-1 white mice. The primary cell cultures obtained were placed into 96-well black flat-bottom plates (Greiner, Monroe, NC, USA) in 0.2 mL of a 199-culture medium (ThermoFisher Scientific, Waltham, MA, USA) per well: 10,000 cells per well for splenocytes, and 20,000 cell/well for Ehrlich’s adenocarcinoma primary cell culture. Ginsenoside Rh2 added at 1.5 and 15 µM, either alone or in combination with 10 µg/mL DOX (25 µM). N = 6 for each condition. Cell cultures were incubated for 3 h at 37 °C. Then, 10 µM DCFDA (as a DMSO stock) was added directly into the incubation medium, cells were placed into a closed thermal shaker (ST-3L, ELMI) for 30 min at 37 °C. Fluorescent measurements were performed on a Fluoroscan Ascent FL plate reader (Labsystems, Philadelphia, PA, USA) with excitation at 485 nm and emission at 538 nm. 

### 4.5. Adhesion Test

Primary cultures of Ehrlich adenocarcinoma cells were derived from 7-day-old tumor re-transplanted into CD-1 mice. Cells were collected by centrifugation, triple-washed with 1% PBS, and plated into a 96-well plate with a cell density of 1.0 × 10^6^ per well (190 µL volume, MEM medium supplemented with penicillin (100 U/mL) and streptomycin (100 mg/mL). Rh2 (stock solution in water: ethanol 100:1) and DOX (stock solution made in water) individually or in combination 1:1 were added to a final concentration within the range 1–25 µM. Plates were incubated for 24 h at 37 °C (5% CO_2_). Then, the wells were washed with cold 1% PBS to remove floating (dead) cells, the live cells were dried and fixed in 96% ethanol. Cell fixation resulted in the cell death, and dead cells were visualized with 0.8% solution of trypan blue. Cell adhesion was estimated microscopically, and then quantified spectrophotometrically (540 nm) after cell lysis with SDS. Results were normalized to the absorbance of untreated cells taken as a 100% control. 

### 4.6. Animals and Ethics Approval 

All in vivo experiments were performed using CD-1 (male) and BALB/C (female) albino mice weighing 25 ± 2 g, 8 weeks old, originally obtained from the vivarium of the Institute of Bioorganic Chemistry, Pushchino, Russia and outbred in G.B. Elyakov Pacific Institute of Bioorganic Chemistry, Far Eastern Branch, Russian Academy of Sciences, Prospect 100-Years of Vladivostok, 159, Vladivostok 690022, Russia. All animals were acclimatized for two weeks and were housed in plastic cages in a standard animal facility under controlled environmental conditions at a temperature of 22 ± 2 °C and a controlled humidity of 50% and 12 h light–dark cycle, with balanced diet for laboratory animals in accordance with the Russian GOST P 50258-92 protocol and water ad libitum.

All animal experiments complied with the ARRIVE guidelines and accorded with the European Convention for the Protection of Vertebrate Animals, Directives 86/609/EEC (Council of Europe European Convention for the Protection of Vertebrate Animals used for Experimental and Other Scientific Purposes. Strasbourg: 1986, Accessed 28 August 2018) 18.III.1986. Council of Europe, ETS No. 123, European Convention for the humane methods for the animal welfare and maintenance (Directive 2010/63/EU on the protection of animals used for scientific purposes EN. Official Journal of the European Union, L 276/33-276/79 (20 October 2010)), the National Standard of the Russian Federation R 53434-2009 ‘Good Laboratory Practice’ (National state standard GOST P 53434-2009, the Russian Federation standard ‘The principles of Good Laboratory Practice’ approved and put into effect by the Order of the Federal Agency for Technical Regulation and Metrology of 2 December 2009, No 544) and permitted by Animal Ethics Committee of Elyakov Pacific Institute of Bioorganic Chemistry (protocol number, 02/21 and date of approval, 26 May 2021). 

### 4.7. Murine Model of Solid Ehrlich’s Adenocarcinoma 

#### 4.7.1. Variant A (Delayed Treatment—After Tumor Formation) 

BALB/C albino mice were inoculated with 0.2 mL containing 1.5 × 10^6^ viable Ehrlich’s adenocarcinoma cells/mice (cells were derived from 7-day-old tumor re-transplanted into CD-1 mice) in the left hind limb (thigh) subcutaneously. Treatment started when the primary tumor reached a size of 57–60 mm^3^. The animals were randomized and divided into 4 groups (*n* = 7 per group): Control (-), a negative control; DOX, Doxorubicin alone (0.25 mg/kg); Rh2, Rh2 alone (10 mg/kg); DOX + Rh2, combined DOX and Rh2 treatment (0.25 mg/kg and 10 mg/kg). All tested remedies and control (NaCl) solution were injected i.p. in a 0.5 mL volume. The treatment continued for 5 consecutive days. Tumor volumes were measured starting on the 10th day after tumor induction every 4 days for a period of 17 days. Tumor sizes were measured using a digital caliper and tumor volumes calculated as: V = π/6 × L × W × H, where V was tumor volume; L, length; W, width, and H, height. On the 22nd day after tumor induction, the experimental animals were euthanized by using the CO_2_ chamber, tumor masses removed and photographed.

#### 4.7.2. Variant B (Early—Next Day Treatment)

BALB/C albino mice were inoculated with 0.2 mL containing 1.5 × 10^6^ viable Ehrlich’s adenocarcinoma cells/mice (cells were derived from 7-day-old tumor re-transplanted into CD-1 mice) in the left hind limb (thigh) via intramuscular injection. The animals were randomized and divided into 5 groups (*n* = 7 per group): Control I, intact; Control N, a negative control (no treatment); DOX, Doxorubicin alone (0.25 mg/kg); Rh2, Rh2 alone (10 mg/kg); DOX + Rh2, combined DOX and Rh2 treatment (0.25 mg/kg and 10 mg/kg).Treatment started the next day (24 h after inoculation) and continued for 5 consecutive days. Tumor growth was assessed as above for variant A. On the 17th day after tumor inoculation, the experimental animals were euthanized by using the CO_2_ chamber, tumor masses removed and photographed.

### 4.8. Murine Model of Ascites Ehrlich’s Adenocarcinoma 

CD-1 albino mice were inoculated intraperitoneally (i.p.) with 0.5 mL PBS containing 3 × 106 cells of Ehrlich’s adenocarcinoma (cells were derived from 7-day-old tumor re-transplanted into CD-1 mice) and randomly subdivided into 4 groups (*n* = 8): no treatment; DOX alone; Rh2 alone; and DOX + Rh2 combination. Four control groups with no drugs, DOX or Rh2 alone, and DOX + Rh2 were tested in parallel with the intraperitoneal tumor inoculated animals. The selected doses of DOX and Rh2 and the administration regime did not induce toxicity or shortening of the healthy mice lifespan (results not shown). Treatment was started 24 h after tumor inoculation and continued for 5 consecutive days. DOX was injected i.p. in a dose of 0.5 mg/kg in 0.5 mL PBS. Rh2 in was i.p. injected in a dose 10 mg/kg in 0.5 mL water-ethanol solution (1:50 *v*/*v*) 1 h after DOX injection. Survival was followed for 100 days after the treatment completion.

### 4.9. Computer Modeling 

Docking experiments were performed using the CDOCKER algorithm as implemented in the Discovery studio 2.5 software suite (Accelrys, San Diego, CA, USA), followed by force field minimization and binding energy calculations using the Keap1 BTB-domain crystal structure with the bound bardoxolone (4CXT.pdb) as a template structure. Structures of bardoxolone, and ginsenosides Rh2-(3β,12β)-12,20-Dihydroxydammar-24-en-3-yl-β-D-glucopyranoside, and 20S-protopanaxadiol-Dammar-24-ene-3β,12β,20-triol, were imported and minimized using ‘Prepare ligands’ protocol after adding hydrogen bonds. Force field minimization was carried out using the molecular mechanics algorithm CHARMm as implemented in Discovery Studio 2.5.

### 4.10. Statistical Analysis

All in vitro assays were performed at least in triplicate, and presented as mean ± SE. Kaplan–Meier analysis for the survival curves in the animal treatment study was carried out using the log-rank test (Mantel–Cox test). Statistical analysis was conducted using SPSS Statistics v.26 (IBM). *p* < 0.05 was considered statistically significant. Additional statistical analyses were performed using the Prism software (GraphPad, San Diego, CA, USA).

## 5. Conclusions

As shown in this work, DOX + Rh2 co-treatment has a dramatic effect on tumor formation from inoculated tumor cells. The combined treatment boosts their individual effects. The profound effect of co-treatment is clearly seen in the ascites model, where no tumor is formed at all in the DOX + Rh2 co-treatment group. A plausible mechanism of their combined treatment is likely based on Rh2 targeting major players responsible for tumor adhesion, which is supported by DOX-induced nuclear damage to impair the cell ability to re-synthesize these proteins and enzymes executing cancer cell attachment and invasion. Based on the in vivo results, DOX + Rh2 co-treatment may be promising as preventive chemotherapy after solid tumor removal.

## Figures and Tables

**Figure 1 molecules-27-00628-f001:**
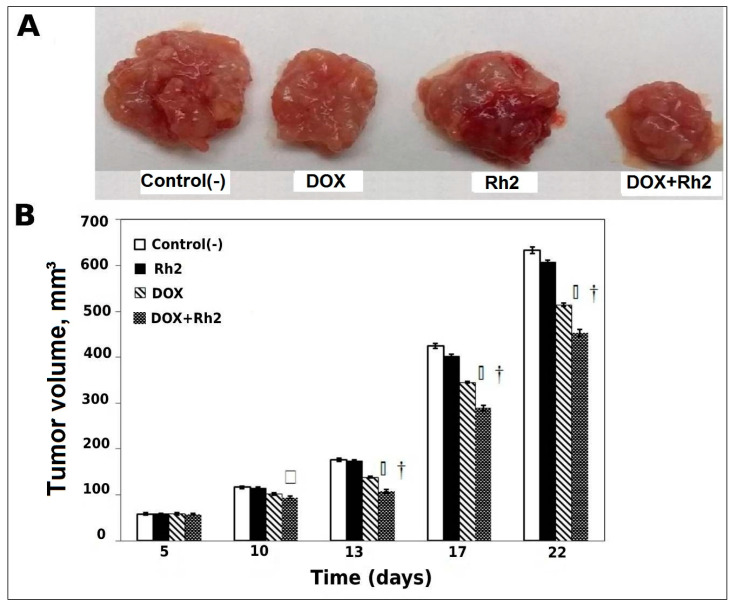
Result of delayed treatment on tumor growth (Variant A). (**A**): Comparison of tumor sizes at the termination (22nd day after tumor induction). Control(-)—a negative control; DOX—Doxorubicin alone (0.25 mg/kg); Rh2—Rh2 alone (10 mg/kg); DOX + Rh2—combined DOX and Rh2 treatment (0.25 mg/kg and 10 mg/kg). (**B**): Average tumor volume at 5–22 days post-induction. □—Between-treatment ANOVA at each concentration, sig. < 0.002. †—Pairwise post hoc *p* < 0.001; DOX or Rh2 + DOX versus Control, or DOX versus Rh2 + DOX.

**Figure 2 molecules-27-00628-f002:**
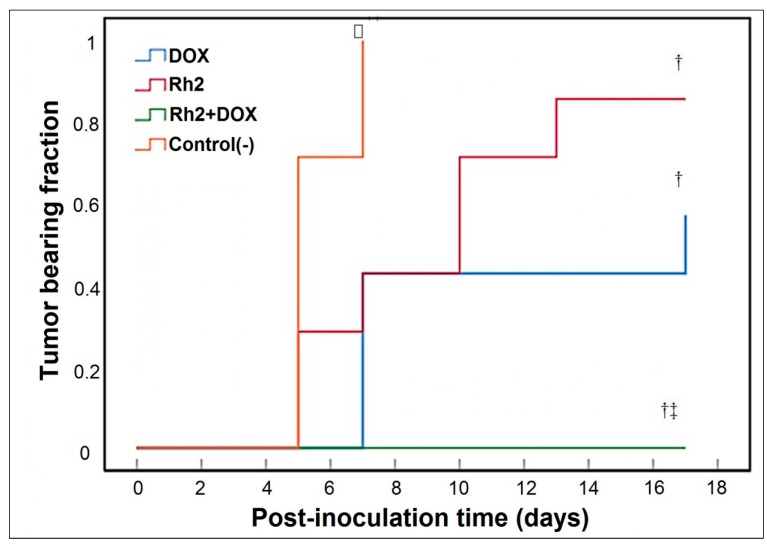
Result of delayed treatment on tumor growth (Variant B, early treatment protocol). Time-course of measurable tumor development (Kaplan–Meier analysis; plotted as one minus event versus time) in animal groups with individual and combined treatment. Control(-)—non-treated animals, used as negative control. DOX—Doxorubicin alone (0.25 mg/kg); Rh2—Rh2 alone (10 mg/kg); DOX + Rh2—combined DOX and Rh2 treatment (0.25 mg/kg and 10 mg/kg). □ Log-rank (Mantel-Cox), *p* < 10^−9^. † Pairwise post hoc log-rank versus control, *p* < 0.02. ‡ Pairwise post hoc log-rank versus Dox or Rh2, *p* < 0.02.

**Figure 3 molecules-27-00628-f003:**
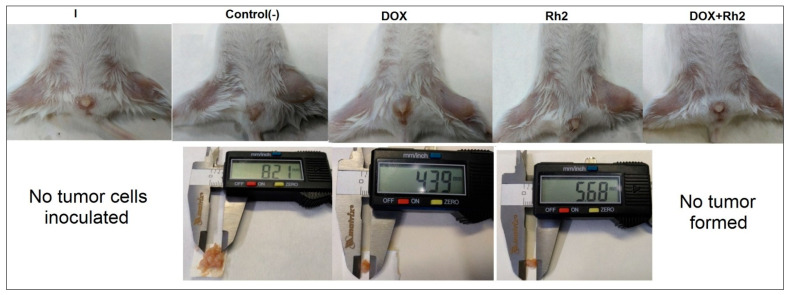
Tumor formation by day 17 (Variant A, early treatment protocol) and its size in non-treated group (N), DOX group, and Rh2 group. No tumor in the intact animals (I), or in DOX + Rh2-treated animals.

**Figure 4 molecules-27-00628-f004:**
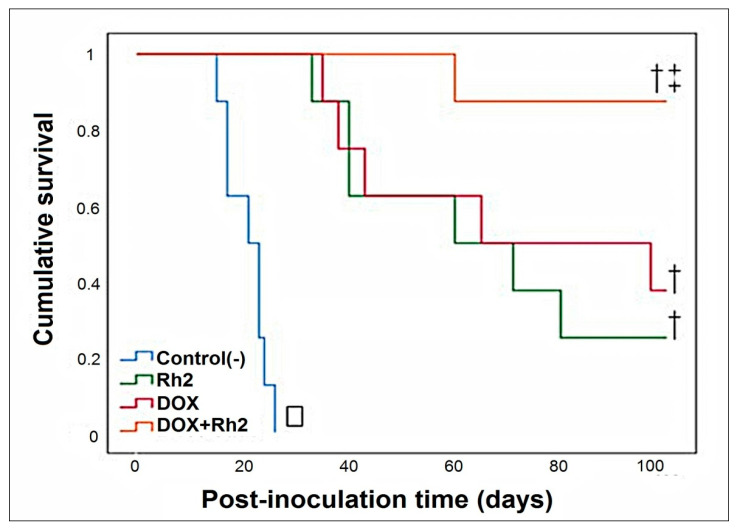
The enhanced effect of DOX + Rh2 co-treatment on ascites adenocarcinoma. Kaplan–Meier survival plot of tumor-inoculated animals treated with DOX or Rh2 alone, and DOX + Rh2 combination. Statistical analysis for the survival curves was carried out using the log-rank test. *p* < 0.05 was considered statistically significant. □ Log-rank (Mantel-Cox) *p* < 10^−4^. † Pairwise post hoc versus control (Mantel-Cox), *p* < 0.05. ‡ Pairwise post hoc versus Dox or Rh2 (Mantel-Cox), *p* < 0.00005.

**Figure 5 molecules-27-00628-f005:**
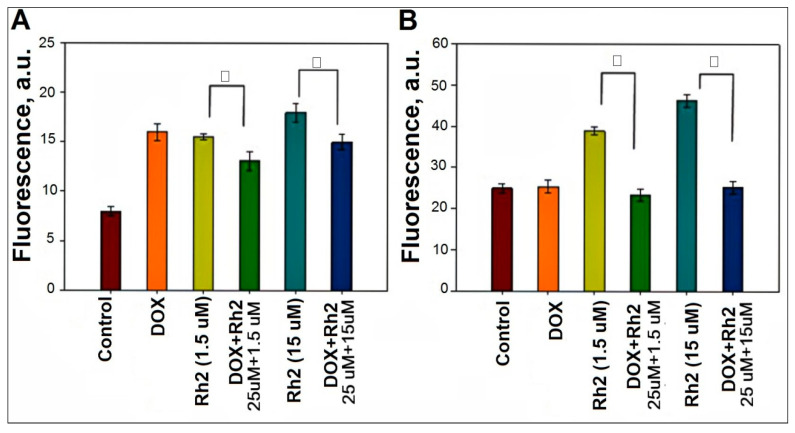
Rh2-induced ROS production assayed by DCFDA fluorescence 3 h after drug treatment. (**A**) adenocarcinoma cells and (**B**) splenocytes were treated with DOX (25 μM) and Rh2 (1.5 and 15 μM). The cell density of adenocarcinoma cells was 20,000 cell/well, and 10,000 cell/well for splenocytes. (N = 6). □ Pairwise comparison as indicated, *p* < 0.01.

**Figure 6 molecules-27-00628-f006:**
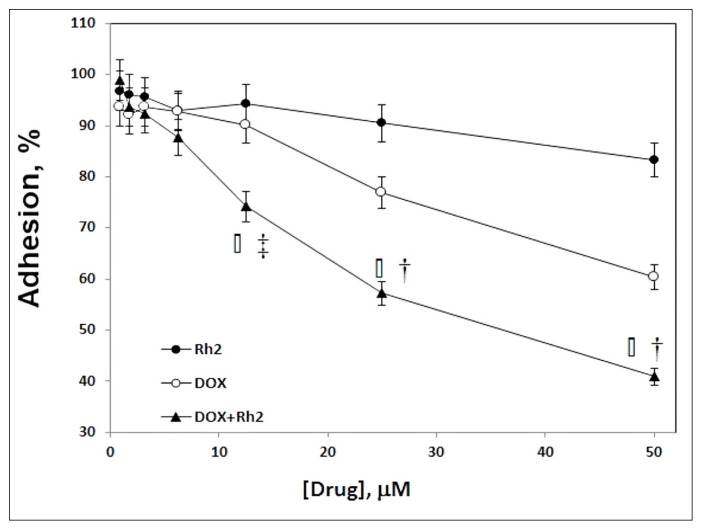
Effect of DOX + Rh2 co-treatment on adenocarcinoma cell adhesion. Evaluation of primary Ehrlich adenocarcinoma cell attachment in the presence of Rh2 and DOX taken individually or as a 1:1 mix upon 24 h incubation. Test performed in triplicate, cell count without treatment taken as 100%, results shown as mean ± SE. (Protocol details under Materials and Methods). *□* Between-treatment ANOVA at same concentration sig, *p* < 0.003. † Pairwise post hoc *p* < 0.001, all pairs. ‡ Pairwise post hoc *p* < 0.001, DOX + Rh2 versus single treatment.

**Figure 7 molecules-27-00628-f007:**
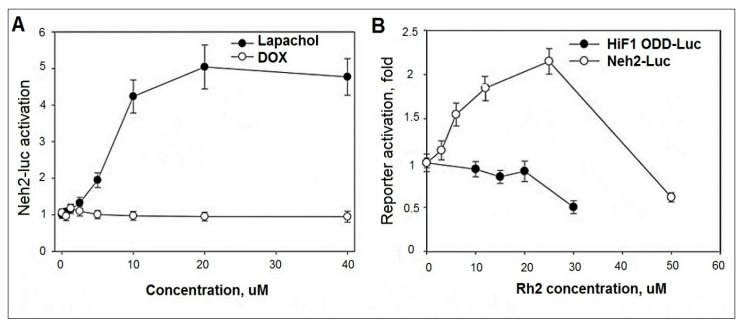
Rh2 as an activator of Neh2-luc reporter. (**A**). Dox does not activate Neh2-luc reporter whereas a well-known ROS producer, beta-lapachone does. (**B**). Rh2 activates Neh2-luc but not HIF1 ODD-luc reporter (3 h incubation). Results shown as mean ± SD (*n* = 3).

**Figure 8 molecules-27-00628-f008:**
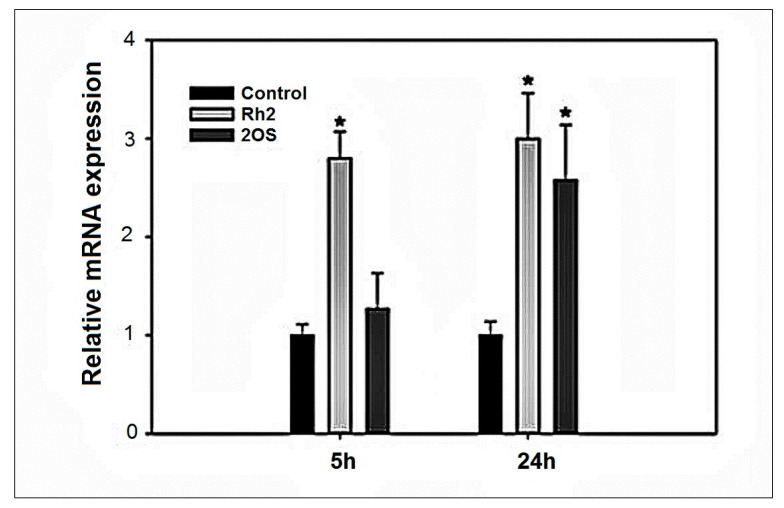
Real-time PCR for HMOX1 expression. HMOX1 expression upon treatment with 25 μM Rh2 and 20S-protopanaxadiol at 5 h and 24 h incubation. Results shown as mean ± SEM (*n* = 3). * *p* < 0.05 versus control.

**Figure 9 molecules-27-00628-f009:**
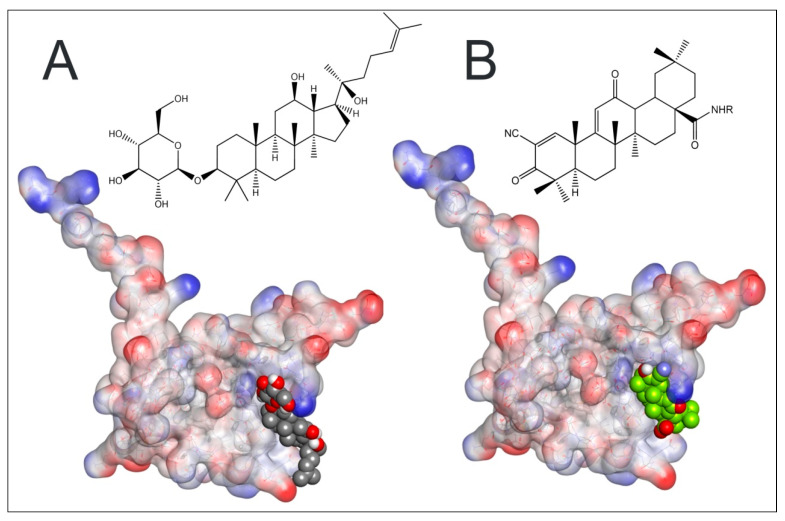
Docking of Rh2 (**A**) in comparison to bardoxolone (**B**) in the crystal structure of Keap1 BTB-domain (4CXT.pdb).

**Figure 10 molecules-27-00628-f010:**
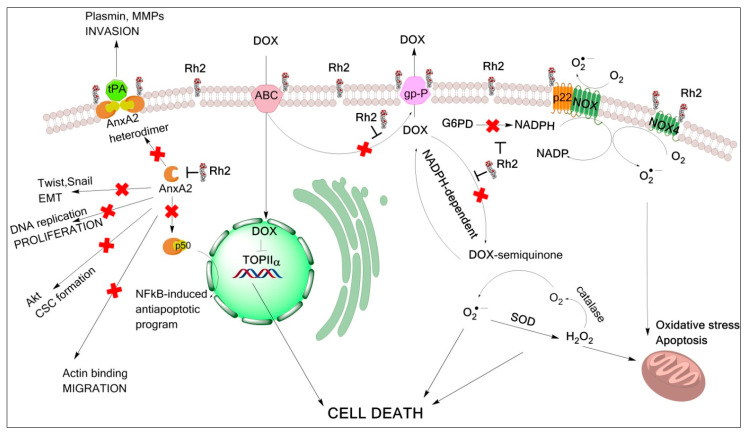
Molecular mechanisms underlying benefits of Rh2-DOX co-treatment in cancer. Major pathway for DOX anti-tumor effect is presented by DOX inhibition of topoisomerase IIa (TOP IIa) shown in the center. Internalization of lipid rafts with Rh2 increases membrane permeability for DOX, which enters the cell through ABC-type transporters. Rh2 inhibits P-glycoprotein (gp-P) which lowers DOX intracellular concentration and is responsible for development of drug resistance. DOX side effects may originate from ROS-production which is NADPH-dependent, and Rh2 inhibits NADPH-dependent DOX redox cycling and possibly glucose-6-phosphate dehydrogenase (G6PD), by analogy with structurally similar enzyme inhibitors. Antimetastatic and anti-apoptotic pathways (left) are inhibited due to Rh2 interaction with annexin A2 (AnxA2). Abbreviations: SOD—superoxide dismutase; NOX—NADPH oxidase; p22—coactivator of NOX; p50—subunit of NF-κB transcription factor, tPA—tissue plasminogen activator, MMP—matrix metalloproteinase, CSC—cancer stem cells, EMT—epithelial–mesenchymal transformation.

## Data Availability

Not applicable.

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
