# Peer review of "Probable Mechanisms of Doxorubicin Antitumor Activity Enhancement by Ginsenoside Rh2"

_molecules, 2022, doi:10.3390/molecules27030628_

Round 1
Reviewer 1 Report
Probable mechanisms of doxorubicin antitumor activity en- 2 hancement by ginsenoside Rh2
In this article (Probable mechanisms of doxorubicin antitumor activity en- 2 hancement by ginsenoside Rh2), it was investigated how Rh2 increases the efficacy of DOX, used in the treatment of cancer, finding that there is a synergistic effect between the ginsen component and the therapeutic compound.
It is a methodologically very well conducted research and above all very interesting and applied, however I have some observations to make.
Figure 1. Results of delay treatment. Because it was decided to measure and compare the tumor size of the treatments control (-), Rh2, DOX y DOX + Rh2?, for 22 days post induction?. What would happen if more time was allowed?
Page 7. Lines 230-231. The authors wrote: Ginseng extract and its components are known to activate Nrf2, although the exact mechanism of activation is still controversial [14–19]. Could the authors point out, relying on this research, what they found? emphasizing their results on this, perhaps in conclusions.
Lines 231-234. Could the authors explain in a better way what they found that has not been previously reported?
Page 10. Lines 315-318. Could the authors explain better why they say that the model of Rh2 Ginsenoside action may explain the observed effect on tumor shrinking for DOX+.Rh2?
Page 12. Line 395. The synthesis of Ginsenoside Rh2 (20(S)-D-glucopyranoside protopanaxadiol) was published 15 years ago, I believe that for this article the authors should provide data on the characterization of this compound, for example purity.
Author Response
Thank you for your attention to our article and your review! You will find our answers in the attachment, please see the attachment

Reviewer 2 Report
The reviewed paper concerns improved effectiveness of anthracycline doxorubicin (DOX) and ginsenoside (Rh2) combination compared to their individual action in a solid tumor model of Ehrlich's adenocarcinoma. The authors also attempted to elucidate the mechanisms of Rh2's biological action using several cell-based molecular approaches.
In view of the increasing incidence of neoplastic diseases, the research undertaken is of great cognitive and practical importance. The experimental part is well planned, the results discussed correctly, and the discussion is the logical argument.
Please note the following:
Line 61: Please give the Latin name of the plant.
Lines 103-104: In the figure caption, the term "DOX + Rh2" appears, then you change the order of the names of the chemical compounds and give the term "combined Rh2 and DOX treatment". Then, in parentheses, you give the dose of individual chemical compounds. So it is unclear which chemical a given dose applies to. To be logical, it is advisable to follow the same order of the chemical compounds.
Lines 100-106: In Figure 1B, some geometric shapes appear above the bars in the graph. The caption of the figure only explains what the figure † means. Please explain the meaning of the other figures.
Line 104: What does the figure “⁎ „mean? No such figure appears in figure 1.
Figure 1, Figure 2, Figure 4, Figure 5, Figure 6: This is true for most of the drawings appearing in the article. What does the rectangle appearing in the pictures mean. Failure to explain the meaning of this geometric figure is somewhat misleading.
Line 124: Same note as for lines 103-104.
Figure 3: please move the captions over the drawings with the mouse so that they are exactly above the drawing they refer to.
Figure 5: Caption under the bar graph "DOX + Rh2 (1.5 µM) and DOX + Rh2 (15 µM). Which compound is in this concentration?
Lines 501 and 515: Same note as for lines 103-104.
Line 544: “in vitro” in italics
Author Response

(The authors gave the same response as above.)
